# An Investigation on the Use of Au@SiO_2_@Au Nanomatryoshkas as Gap-Enhanced Raman Tags

**DOI:** 10.3390/nano13212893

**Published:** 2023-11-01

**Authors:** Brinton King Eldridge, Saghar Gomrok, James W. Barr, Elise Anne Chaffin, Lauren Fielding, Christian Sachs, Katie Stickels, Paiton Williams, Yongmei Wang

**Affiliations:** 1Department of Chemistry, University of Memphis, Memphis, TN 38152, USA; bldrdge1@memphis.edu (B.K.E.); sgomrok@memphis.edu (S.G.); 2Department of Biological, Physical, and Human Sciences, Freed-Hardeman University, Henderson, TN 38340, USA; jbarr@fhu.edu (J.W.B.); echaffin@fhu.edu (E.A.C.); lfieldin@uab.edu (L.F.); 1christian9953@gmail.com (C.S.); katie.stickels@students.fhu.edu (K.S.); paitonw6@gmail.com (P.W.)

**Keywords:** discrete dipole approximation, gap-enhanced raman tag, localized surface plasmon resonance, multilayer, nanoparticle, Raman, surface enhanced Raman scattering, nanomatryoshka

## Abstract

Gap-enhanced Raman tags are a new type of optical probe that have wide applications in sensing and detection. A gap-enhanced Raman tag is prepared by embedding Raman molecules inside a gap between two plasmonic metals such as an Au core and Au shell. Even though placing Raman molecules beneath an Au shell seems counter-intuitive, it has been shown that such systems produce a stronger surface-enhanced Raman scattering response due to the strong electric field inside the gap. While the theoretical support of the stronger electric field inside the gap was provided in the literature, a comprehensive understanding of how the electric field inside the gap compares with that of the outer surface of the particle was not readily available. We investigated Au@SiO${}_{2}$@Au nanoparticles with diameters ranging from 35 nm to 70 nm with varying shell (2.5–10 nm) and gap (2.5–15 nm) thicknesses and obtained both far-field and near-field spectra. The extinction spectra from these particles always have two peaks. The low-energy peak redshifts with the decreasing shell thickness. However, when the gap thickness decreases, the low-energy peaks first blueshift and then redshift, producing a C-shape in the peak position. For every system we investigated, the near-field enhancement spectra were stronger inside the gap than on the outer surface of the nanoparticle. We find that a thin shell combined with a thin gap will produce the greatest near-field enhancement inside the gap. Our work fills the knowledge gap between the exciting potential applications of gap-enhanced Raman tags and the fundamental knowledge of enhancement provided by the gap.

## 1. Introduction

When a metal nanoparticle (NP) is illuminated by electromagnetic radiation, the electron cloud begins to oscillate collectively with the incident electric field, resulting in an enhanced electric field on the surface of the metal [1]. The strength of these oscillations are dependent on an NP’s characteristics, including geometry, size, metal composition, environment, and more [2,3]. This is known as an NP’s localized surface plasmon resonance (LSPR) and it is the driving force of many different applications in nanotechnology. This knowledge of how an NP system interacts with light is ubiquitously useful to many fields including photocatalysis [4,5,6], drug delivery [7], tumor diagnosis and therapy [8], and nanoscale optical parametric amplification [9]. Many such applications utilize the enhanced electric field on the surface of the metals to produce optically sensitive systems.

One such application includes Raman molecules like pyridine [10], dye-tagged DNA strands [11], and Raman-active polymers [12] on the NP’s surface. The Raman signal is significantly enhanced compare to the signals from Raman molecules in the absence of the surface. This phenomenon was coined as surface-enhanced Raman scattering (SERS) [13], a field pioneered by Richard P. Van Duyne in the late 1970s [14]. It was not until the 2000s that Van Duyne and his colleagues began to study Raman molecules adsorbed on the surface of metal NPs, which led to a better understanding of the mechanism of SERS. It was found that Raman molecules on the surface of an NP can have enhanced Raman signals from six to nine orders of magnitude higher than the isolated Raman molecule counterparts [15,16,17] largely due to the enhanced electric field on the surface of the NP [15]. This enhancement provides an increased detection sensitivity such that signals from an individual Raman molecule [18,19] and the differences in the orientation of the molecules on the metal surface [13] can be detected and differentiated.

The combination of Raman-active molecules with metallic NPs, also known as a SERS tag, has been used in many fields as optical nanoprobes [10,20,21]. Coating a SERS tag with a biologically safe layer like bovine serum albumin or polyethylene glycol [21] would allow for detection in biological systems in vivo or in vitro. Attaching antibodies to the surface of the SERS tag would allow for specific antigen targeting and detection. For anything presenting these antigens, the SERS tag would bind and the Raman response of that region would be quickly identified by the presence of the SERS tag. This process and similar derivatives were used for diagnosis during the COVID-19 pandemic [22,23]. In addition, other SERS tag constructs have been shown to differentiate breast cancer cells from healthy cells [24]. Thus, the investigation of optimized SERS tags can advance future bio-sensing methods.

Typically for SERS tags, Raman molecules are placed on the surface of the NPs. However, Lim et al. [11] reported a study where Raman molecules were placed inside the gap between an Au core and Au shell (many referred to this structure as a nanomatryoshka (NM), an allusion to the Russian word for nesting doll). One would expect that by embedding the Raman molecules inside the gap and thereby protected them by an Au shell, the Raman signals would not be detected. Surprisingly, they have shown that such a system can give rise to highly sensitive and strong SERS spectra because of the electric field enhancement in the gap. In addition to the gap enhancement, the placement of Raman molecules in the gap of an NM has several advantages over the traditional surface Raman molecule placement [17,25]: protection from desorption; a stable response not disrupted by aggregation; linear correlation between GERT concentration and Raman intensity; prevention of photobleaching; and the ability to incorporate different Raman molecules into a multilayered NM. The review by Khlebtsov et al. [17] discussed multiple means by which to synthesize a GERT NM for the purpose of controlling the size of each layer, including the synthesis of a Au@SiO${}_{2}$@Au NM [26]. Thus, the production and adjustment of a GERT NM is well documented and reproducible.

Further studies on these gap-enhanced Raman tags (GERTs) [25] focused on the determination of optimal NM geometries for SERS properties [12,25,26,27,28,29,30]. Khlebtsov and Khlebtsov [29] used Mie theory and examined the electric field enhancement inside the gap of such NMs. By making use of Mie theory, they obtained the surface-averaged electric field inside the gap and examined how the gap and shell thickness impacted the surface-averaged electric field enhancement. They have shown that as the core size increases, the peak of electric field enhancement spectra redshifts, and the peak intensity maximizes at a certain core size. Similarly, they have investigated the impact of shell thickness on the electric field enhancement in the gap. Their studies have provided clear evidence that the electric field inside the gap is strongly enhanced. However, they have examined only a very thin gap (1 nm) as their studies were largely motivated by experimental studies in Lim et al. [11] Additionally, they have presented the averaged electric field inside the gap, the electric field decay pattern throughout the NM, but only small notes on the angular electric field fluctuation in the gap itself. Thus, the results of our work can serve to add clarity to and additional information on the NM.

The earlier studies on NM systems mainly present on the optimal shape parameters in a certain tested range; however, there are few studies on the electric field enhancement inside the gap to the degree of detail we provide. Here, we investigate Au@SiO${}_{2}$@Au NMs using the discrete dipole approximation (DDA). The methods used in the DDA approach enables the researcher to examine a multitude of descriptive attributes of the system being investigated, an advantage we make use of in this paper. Specifically, we present details of the far-field and near-field response of many tested NM sizes as well as parameters such as the electric field distributions inside the NM and average electric field throughout the NM in order to provide a better understanding of the electric field that interacts within the GERT.

## 2. Methodology

For a homogeneous sphere, the solution to the Maxwell equations by Gustav Mie describes the scattering of an incident electromagnetic plane wave exactly. Since the formulation of Mie Theory, there have been multiple solutions augmenting Mie Theory for other structures. Included in this is the development of a multilayered sphere Mie Theory [31], and computational algorithms of this method are available [32]. While Mie theory can investigate the near-field of such NMs, it is quite challenging to obtain the electric field at all spatial locations within the NM. Thus, other methods are better suited.

This study uses the Discrete Dipole Approximation (DDA) to study Au@SiO${}_{2}$@Au NMs (on the order of ∼50 nm in diameter). The DDA method is a self-consistent field approach that is used to obtain information about the induced dipoles of the NP at every point due to incident electromagnetic waves. Thus, this approach is limited to classical electromagnetic physics, but many studies have shown that the computational results agree with experimental values [33]. For particles of these sizes, it has been reported that a quantum mechanical description agrees with classical approaches [27,34]. Comparable computational methods like Finite Element Method (FEM) and Finite Difference Time Domain (FDTD) do not scale well in time and accuracy for differently sized systems [35,36,37]. A comparison of our DDA results with the multilayer Mie theory by Yang et al. [32] can be found in Figure 1, and the DDA and Mie calculations align almost perfectly. A previous work from our group compared the near-field spectra obtained with DDA to that from Mie theory for spherical Au NPs [38] and found good agreement. Thus, the use of DDA provides an accurate and time-effective simulations of the optical properties of NMs.

A freely available application of the DDA is DDSCAT 7.3, which defines a system as a matrix of surveyed points acting as dipoles [39,40,41,42]. An incident light wave with an electric field $E_{\mathrm{inc}}$ causes changes to each dipole, which in turn change other dipoles; this process is iteratively computed until values converge within a predefined accuracy. As input, the DDSCAT method requires dielectric response data of the material to determine the material’s polarizability (α). Experimentally determined dielectric values from by Johnson and Christy [43] were used for Au and values by Malitson [44] were used for SiO${}_{2}$. An adjustment from the bulk dielectric values to account for either the size [45] or shape of the nanosystem has been found to be unnecessary. Finally, the regular three-dimensional matrix of dipoles is then partitioned into segments of the aforementioned material’s polarizability, as described by a shape file fed into the program’s initialization.

At its simplest, DDSCAT evaluates each induced dipole $P_{j}$ as the polarizability of point *j* ($\alpha_{j}$) is multiplied by the sum between the incident electric field at point *j* ($E_{\mathrm{inc},j}$) and the electric field from every other point *i* on point *j* ($E_{i,j}$): (1)$$P_{j}=\alpha_{j}\left(E_{\mathrm{inc},j}+\sum_{i}E_{i,j}\right).$$
This computation results in a matrix of $P_{j}$ at each surveyed point (indexed with *j*). Despite the discrete nature of the matrix, any point’s $P_{j}$ can be estimated by linear interpolation using neighboring dipoles. Once all $P_{j}$ values are calculated, we find the far-field cross sections: (2)Cext=4πk|Einc|2∑j=1NImEinc,j*·PjCabs=4πk|Einc|2∑j=1NImPj·(αj−1)*Pj*−23k3|Pj|2
where *k* is the incident light wave-vector ($k=\frac{2\pi }{\lambda }$), *N* is the total number of dipoles, * indicates the complex conjugate, and $\alpha_{j}$ is defined by the Filtered Coupled Dipole (FLTRCD) method, which uses the dielectric response data [42]. Lastly, to compare extinction and absorption of different systems, we redefine them in terms of the unit-less Q${}_{i}=\frac{C_{i}}{\pi r^{2}}$, where *r* is the radius of the NP. Finally, the relationship Q${}_{\mathrm{ext}}$ = Q${}_{\mathrm{abs}}$ + Q${}_{\mathrm{sca}}$ can be used to find Q${}_{\mathrm{sca}}$.

Moreover, since we have access to all electric fields at the dipoles surveyed by DDSCAT, we have developed a routine in which we obtain all the electric field values of a spherical surface of sites at a certain distance from the center. This way, we are able to calculate the surface-averaged electric field at any distance as well as the distribution of the electric field throughout one spherical surface of points. We note that, even though the electric field is most intense just above the core surface (∼10.4 nm due to our DDSCAT dipole distance), a measurement of 1 nm from the surface behaves more consistently like the analytical Mie Theory solution than any other distance [38]. So, for measurements denoted as “Inner Surface”, we measure the electric field 1 nm above the surface of the Au core and for “Outer Surface”, we measure 1 nm above the outer surface of the Au shell.

## 3. Results and Discussion

### 3.1. Plasmonic Combinations

As the incident electric field interacts with the Au@SiO${}_{2}$@Au nanomatryoshka (NM), it will induce a plasmon resonance of the conduction electrons on the surface of the Au components. For the plasmon of a small, spherical NP, this resonance happens predominately in a dipolar fashion [46]. However, since there are multiple components of this structure, there will be multiple possible plasmon modes. For a nanoshell (NS), which is a combination of a sphere and a cavity in a bulk metal, two plasmonic modes are possible: an asymmetric mode where, for each hemisphere, the charge on the outer surface is the opposite sign of the charge in the inner surface of the shell; and a symmetric mode where, for each hemisphere, the charges on the inner and outer surface of the shell have the same sign [47,48]. The former is naturally at a higher energy than the latter. For the insertion of a spherical NP into an NS to create an NM, combinatorics defines four possible plasmon modes (Figure 2).

The energies of the four possibilities have been analytically predicted using the plasmon hybridization theory (PHT) [28,34,47,48,49,50]. However, the distribution of energies can be explained by the presence or absence of one or more high-energy choices (indicated by the flow chart of Figure 2). For the surface charges of the core and inner shell, coulombic interactions between like charges cause choice 1a to be greater in energy than choice 1b. Secondly, having a different surface charge between the inner and outer shell surfaces causes choice 2a to be greater in energy than choice 2b. This calls back to the plasmon modes of a simple Au shell, where the high-energy mode is thought to have different surface charges on its inner and outer portions [28,47,51]. However, the judgement of which choice causes a greater increase in energy is left to computational evidence. According to surface charge distributions from multiple works [28,30,52], the energy of the plasmonic mode of Figure 2c is shown to be lower than the energy of Figure 2b. Therefore, the coulombic interaction inside the gap causes a greater energy change as compared to the change produced by different surface charges of the inner and outer portion of the shell.

Overall, the plasmonic mode exhibiting both high-energy choices (Figure 2a) is thought to be not physically possible; thus, it does not appear in any spectra (it is not even mentioned in the PHT papers above). Figure 2b is predicted to be in the UV spectrum and in a particular position where the dielectric functions used [43,44] are not measured [50]. The plasmonic modes of Figure 2c,d are optically active in the visible range with the mode exhibiting the two low-energy choices (Figure 2d) being the lowest in energy.

Figure 3 shows a comparison of the extinction efficiencies of an Au NP, an Au NS, and an NM. Efficiency, symbolized by Q, is the respective cross section (extinction, ext; absorption, abs; scattering, sca) divided by the cross-sectional area of the particle. Only one plasmon is optically active for the NS in this wavelength range, which is the lower energy symmetric dipole plasmon. The NS’s asymmetric dipole LSPR resides out of this wavelength range in the UV spectrum [50]. When a Au NP is inserted into a NS to create a NM, the NS peak splits into two: the blueshifted, higher-energy peak on the left we ascribe to the non-bonding plasmon mode (Figure 2c); and the redshfited, lower-energy peak on the right we ascribe to the bonding mode (Figure 2d).

### 3.2. Far-Field Response

To learn about the effect of each component of the NM, we began by fixing both the core and SiO${}_{2}$ gap thicknesses and changing the Au shell thickness. The spectra of the far-field extinction efficiency of each system are found in Figure 4. In addition, absorption and scattering efficiency spectra for all series studied were placed in the Appendix A (Appendix A). These systems are highly absorptive; thus, the extinction and absorption spectra have the same trends. These extinction spectra show the NM-characteristic two peaks: the non-bonding plasmon mode at the short wavelength on the left and the bonding plasmon mode at the longer wavelengths on the right (Figure 4). However, these two peaks do not behave the same when subject to a changing shell thickness.

As the shell gets thinner, the bonding peak redshifts to a lower plasmonic energy. A thinner shell will allow a greater electric field intensity through it, thus increasing the core’s plasmon strength. For the bonding mode, there exists a coulombic attraction between the oppositely charged plasmons of the core and the shell; therefore, the strength of this attractive force lowers the required energy for resonance as the core’s plasmon strength increases. However, the position of the non-bonding mode peak is stationary through the changes indicating no influence of the shell thickness to that mode. In fact, its position being so close to the original Au core peak seen in Figure 3 speaks to the influence of the core on the non-bonding peak and not the shell’s influence.

Next, we investigated the dependence of the SiO${}_{2}$ gap on the NM’s interaction with light. Figure 5 shows the Q${}_{\mathrm{ext}}$ of systems with different SiO${}_{2}$ thicknesses. Again, two peaks are shown: the left non-bonding peak and the right bonding peak. In addition, a C-shape can be seen in bonding mode peak as the SiO${}_{2}$ gap thickness decreases (indicated by the added arrow in Figure 5), which provides some detail about the effect of the gap. It has been reported that after passing through a metal nanolayer, an electric field decays to normal levels within 10−20 nm [28,53]. Thus, at the beginning of the C-shape, the gap is so large that the induced electric field by the shell decays before it can couple with the core, effectively separating the core from the shell. Thus, the system acts like a constant-thickness Au NS system: blueshifting as the overall size decreases (Appendix A). The shell then becomes close enough to the core to plasmonically couple, which causes a redshift as the SiO${}_{2}$ gap decreases. Because this shift is only evident in the bonding mode, we again suspect the cause of this shifting to be the coulombic attraction of the oppositely charged plasmons of the core and shell. The non-bonding mode has a slight blueshift as the gap decreases; this is likely due to the increase in energy of the coulombic repulsion between the similarly charged surfaces of the core and shell.

### 3.3. Near-Field Response

The far-field properties of an NP are important for experimental detection and confirmation of the desired shape and size. Here, the far-field spectra has been used to assign plasmonic modes; however, for SERS applications, understanding of the near-field of the NP is far more important. For the near-field enhancement (NFE), a measurement of local electric field strength, we use the following equation: (3)$$\mathrm{NFE}=\frac{\langle {\left|E\right|}^{2}\rangle }{|E_{\mathrm{inc}}{|}^{2}}$$
where $|E_{\mathrm{inc}}|$ is the incident electric field and $\langle {\left|E\right|}^{2}\rangle$ is the average of the square of the electric field intensity surrounding the particle at a certain distance from the center.

Figure 6 presents the near-field response of the changing shell series.The influence of the shell can be observed from NFE${}_{i}$ (Figure 6a), NFE${}_{o}$, (Figure 6b) and their ratio NFE${}_{i}$/NFE${}_{o}$ (Figure 6c). The NFE${}_{i}$ would correspond the SERS enhancement of Raman molecules placed inside the gap, and NFE${}_{o}$ would be the enhancement of the Raman molecules that are placed on the outer surface. The ratio of the two, NFE${}_{i}$/NFE${}_{o}$, reveals enhancement gained by placing a Raman molecule inside the gap compared to at the outer surface.

The NFE${}_{i}$ intensity at the bonding peak remains high through the changes in shell thickness with a slight increase as shell thickness is decreased (Figure 6d). Since each experiment shows a high NFE${}_{i}$, the GERT would still be effective despite experimental abnormalities in shell thickness. The NFE${}_{o}$ (Figure 6b) behaves similarly to Q${}_{\mathrm{ext}}$ (Figure 4); this is also true for the gap series (Figure 5 and Figure 7b). This similar behavior can be understood by the calculation for the far-field extinction, which utilizes the sum of all local E intensities. Since there exist significantly more surveyed points at the surface of the shell than the surface of the core, the NFE${}_{o}$ will share more character with Q${}_{\mathrm{ext}}$ than the NFE${}_{i}$. Thus, the NFE${}_{o}$ can be expected to correlate with the trends in Q${}_{\mathrm{ext}}$. For the trend of NFE${}_{o}$, the bonding peak intensity increases with a decrease in shell thickness, which is the opposite trend for the non-bonding peak intensity (Figure 6e). The general comparison between NFE${}_{i}$ and NFE${}_{o}$ in Figure 6c highlights the increase in NFE${}_{o}$ while the NFE${}_{i}$ is relatively stable. However, the greater magnitude of NFE${}_{i}$ as compared to NFE${}_{o}$ for every wavelength provides strong theoretical support for the benefits of gap-enhanced Raman tags.

Near-field spectra for this gap series are shown in Figure 7. There are four observations of the NFE that we want to highlight. First, in every NM, we note that the NFE${}_{i}$ is again significantly greater than the NFE${}_{o}$ (Figure 6a–c and Figure 7a–c). Second, the peak intensity of NFE${}_{i}$ increases with a decrease in gap thickness (Figure 7d). This further suggests the advantage of using GERT, as the enhancement in the gap is significantly greater than the outer surface. In addition, a thinner gap increases this enhancement. As a note, at smaller gap thicknesses (2.5 nm), the calculation by DDA reveals a relatively large increase in NFE${}_{i}$ intensity. Although this change is still in line with the overall trend of the data, it is important to note the possible quantum effects occurring at such small gap sizes, which DDA is not able to account for. Third, for the NFE${}_{o}$, the increasing core-shell coupling allows the non-bonding mode to increase in intensity while the bonding mode significantly decreases in intensity (Figure 7e). Fourth and finally, for Figure 7f, a smaller gap thickness very obviously produces an enhanced electric field inside the gap as compared to that on the outer surface of the NM.

Throughout the analysis of both the gap and shell series, the experimental application is of utmost importance. In comparing the near-field spectra to the far-field spectra in every system examined (including those in Appendix A), we observed that the position of a system’s bonding peak was different between NFE and Q${}_{\mathrm{ext}}$. This peak shift has been seen in previous studies [38]. The peak of NFE corresponds to the maximum SERS enhancement. Ideally, for Raman detections, one should try to maximize Raman signals by using incident laser wavelength corresponding to the peak in NFE. Practically, however, an experimentalist will only be able to easily measure the Q${}_{\mathrm{ext}}$ of an NM. The NFE${}_{i}$ spectra have an intense bonding peak that makes the non-bonding peak hard to pinpoint, so we provide the shifts of the bonding peaks in the NFE${}_{i}$ and NFE${}_{o}$ as they relate to the measurable Q${}_{\mathrm{ext}}$ bonding peak in Figure 8. On average, the NFE${}_{i}$ peak coincides with the Q${}_{\mathrm{ext}}$ peak, but the NFE${}_{o}$ peak is 8 nm redshifted from the Q${}_{\mathrm{ext}}$ peak. The remaining correlation between NFE${}_{i}$ and NFE${}_{o}$ is provided in Appendix A.

### 3.4. Variation of Near-Field Intensity in the Gap

The observed enhancement of the electric field intensity inside the gap is further evidence to support the use of GERTs. The position of Raman molecules in the gap is controlled mainly by the type of Raman molecule deployed in the gap [17]. Usually, this layer of Raman molecules resides just on the surface of the Au core, but it is possible to place the layer at any distance from the core. Hence, we examined the NFE as a function of distance from the center of the NM and found its peak to be at the surface of the core (blue line, Figure 9). We also find this to be another dimension of evidence in support for the use of Raman molecules in the gap since the ${\left|E\right|}^{2}$ is much larger in the gap than the outer surface.

Nevertheless, the NFE in Figure 9 is only an average. It is important to gauge the expected fluctuation inherent in a Raman molecule layer, whether the Raman molecules are on the outside of the sphere or imbedded in the gap. Thus, for a measure of variation in ${\left|E\right|}^{2}$, we used the coefficient of quartile variation (cqv): (4)$$\mathrm{cqv}=\frac{Q_{3}-Q_{1}}{Q_{3}+Q_{1}}.$$
where Q${}_{1}$ and Q${}_{3}$ are the first and third quartiles, respectively. Since cqv operates on quartile information alone, it is less sensitive to outliers and provides the dispersion of a dataset [54]. The green dataset in Figure 9 presents how cqv varies along the radial distance. Through the cqv, we find that the fluctuation in the electric field intensity in the gap is slightly higher compared to the outside of the sphere.

To gain further insight on this variation of the electric fields both in the gap and on the outside of the NM, we present a probabilistic histogram (Figure 10a) and accompanying visualization (Figure 10b) detailing the spread of the electric field intensity for the system in Figure 9 as it changes angularly. This graph plots the probability of a point 1 nm above both the core (inner surface) and shell (outer surface) having a given electric field intensity. So, the tighter peak for the outer surface indicates a lesser distribution of points. Thus, overall, the electric field inside the gap varies greatly, as emphasized by the large right-hand tail in probability and high cqv value. This is in agreement with other studies on smaller NM [29]. In addition to this, the figure shows that the inner surface has significantly more sites that have a higher electric field intensity than the outer surface. This is consistent with the earlier observation that NFE${}_{i}$/NFE${}_{o}>1$ at that wavelength. Additionally, it is consistent with Figure 10b where the electric field intensity at two wavelengths is shown. For all systems tested, this trend is true and can be seen in the box plots of Appendix A.

This variation of the electric field on a Raman molecule layer can cause signal fluctuation, which is not ideal. Particularly, with the embedding of only a few Raman molecules in the gap, the Raman response can vary depending on the light polarization and the orientation of the NM. For an experimental mixture of GERTs, alignment of a Raman molecule with a polarized beam is unlikely; however, with a full layer of Raman molecules in the gap, the signal fluctuation between GERT NMs will be reduced, providing a reliable Raman signal.

## 4. Conclusions

In this study, we first discussed the four possible plasmonic modes of an NM. For the combination of an Au nanoshell (of which there are two possible modes) and a solid Au sphere (of which there is only one mode), four plasmonic modes for an NM are possible. In a far-field Q${}_{\mathrm{ext}}$ spectra one can easily see that the lower energy nanoshell mode combines with the only plasmonic mode of the sphere to create two peaks in the NM response: one constantly around 540 nm denoted as the “non-bonding” peak and one on the red side of the spectrum with a variable peak wavelength denoted as the “bonding” peak. The elaboration of the plasmonic modes at each peak helped create a better understanding of the far-field response of each system.

We investigated the influence of the shell on the system and found that a thinner shell increases the NFE${}_{i}$ peak intensity, but a thicker shell has a greater ratio of NFE${}_{i}$/NFE${}_{o}$. For the influence of the SiO${}_{2}$ gap, we found that at a thickness greater than 10 nm, the shell does not interact fully with the core. Less than 10 nm, the SiO${}_{2}$ gap is small enough to enable coupling between the core and shell, and so the peaks redshift with a decreasing gap thickness. At the smallest measured SiO${}_{2}$ thickness of 2.5 nm, the ratio NFE${}_{i}$/NFE${}_{o}$ is greatest. Overall, we confirm that the NM that will produce the highest NFE${}_{i}$ should have a thin shell and thin gap. Nonetheless, for every system tested, the average NFE${}_{i}$ was always greater than its NFE${}_{o}$ counterpart, reinforcing the advantage of GERTs over traditional SERS NP. For all systems, the NFE${}_{i}$ peak intensity nearly coincide with the bonding peak in the Q${}_{\mathrm{ext}}$.

Finally, we found that the NFE is highest at the surface of the core in each NM, which was not unexpected. However, we also evaluated the electric field 1 nm from the core and 1 nm from the outer surface of the shell and found that the variation in intensity in the gap is greater than the variation of the surface of the shell. The signal strength of the Raman molecules in the gap will depend on their position relative to the electric field polarization if the Raman molecules do not form a full layer, hence bringing a greater signal fluctuation between experimental measurements. However, a full Raman molecule layer will negate the polarization specificity and provide low signal fluctuation between GERT NM in solution.

The current study could be extended through two future investigations. First, we only investigated Au@SiO${}_{2}$@Au NMs. Other GERT geometries are possible to synthesis and have different optical responses [17]. For example, structures such as petal-like Au core@Ag@SERS NMs have shown great results in biomedical applications [55,56]. Secondly, we used DDA to describe the potential SERS enhancement experienced by Raman molecules when placed inside NM, as compared to the SERS enhancement when Raman molecules are placed on the outer side of the surface of the NMs. The nature of DDA uses classical electrodynamics. No quantum effects are included in these calculations. However, at small gap thicknesses less than 2.5 nm, it is likely that quantum effects can play a role. Additionally, the interaction of dye molecules with the plasmonic particle has not been taken into account. The interaction of Raman molecules with the metal surface can be non-trivial, especially for a full layer of tags that share a similar resonant wavelength [57,58,59,60,61]. In order to account for these effects, quantum mechanical studies that address the coupling between Raman molecules and metal NPs are needed and will be the focus of future works.

## Figures and Tables

**Figure 1 nanomaterials-13-02893-f001:**
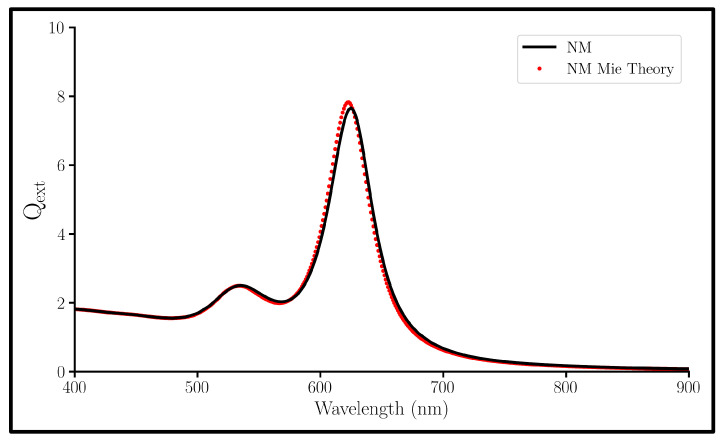
Q${}_{\mathrm{ext}}$ spectra of a 10 nm Au shell, 10 nm SiO${}_{2}$ gap, and a 10 nm radius Au core nanomatryoshka (NM) calculated with DDA (black) and Mie Theory (red).

**Figure 2 nanomaterials-13-02893-f002:**
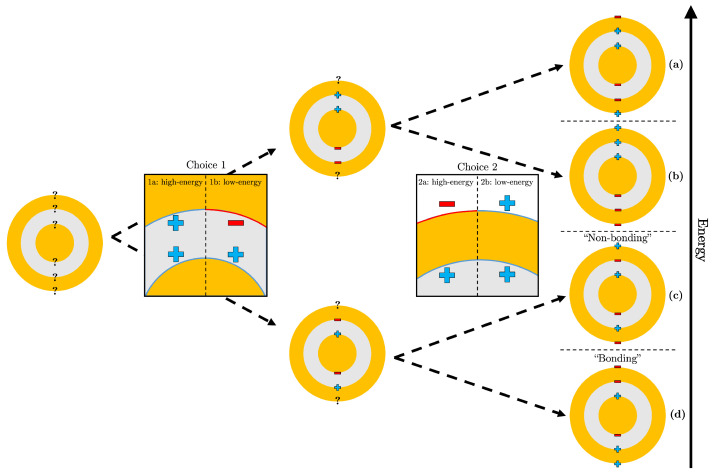
Flow chart detailing different surface charge distributions for the four combinatorial plasmon modes of an Au@SiO${}_{2}$@Au nanomatryoshka where (**a**) is predicted to be the highest energy mode, (**b**) second highest, (**c**) second lowest energy “non-bonding” mode, and (**d**) lowest energy “bonding” mode.

**Figure 3 nanomaterials-13-02893-f003:**
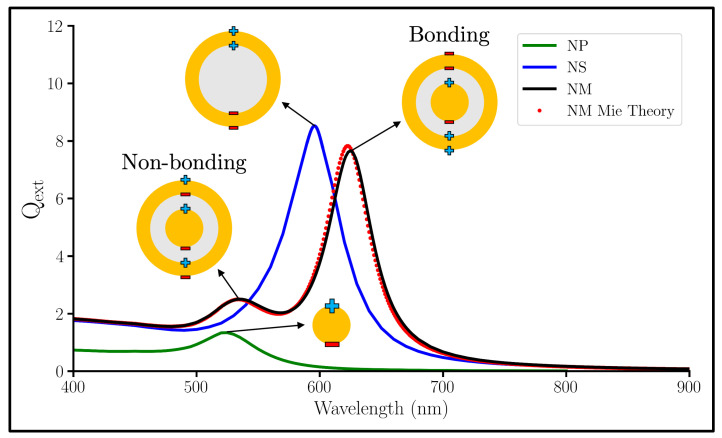
Q${}_{\mathrm{ext}}$ spectra of a 10 nm radius Au NP (green); a 10 nm thick Au nanoshell (NS) with a 20 nm radius SiO${}_{2}$ core (blue); and a 10 nm Au shell, 10 nm SiO${}_{2}$ gap, and a 10 nm radius Au core nanomatryoshka (NM) calculated with DDA (black) and Mie Theory (red).

**Figure 4 nanomaterials-13-02893-f004:**
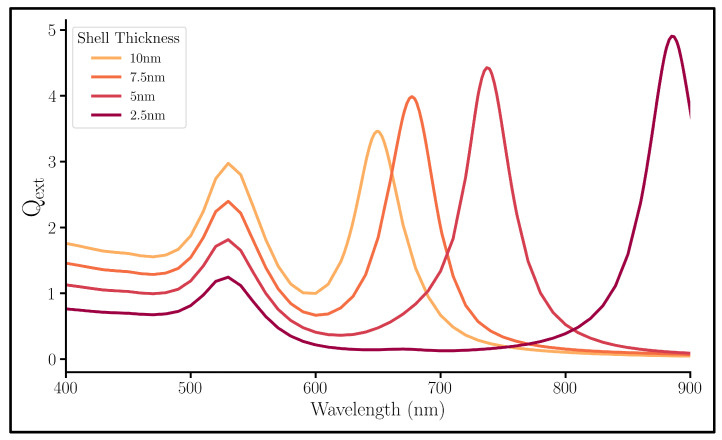
Q${}_{\mathrm{ext}}$ spectra of nanomatryoshkas with a variable Au shell thickness, a fixed 5 nm SiO${}_{2}$ gap, and a fixed 10 nm radius Au core.

**Figure 5 nanomaterials-13-02893-f005:**
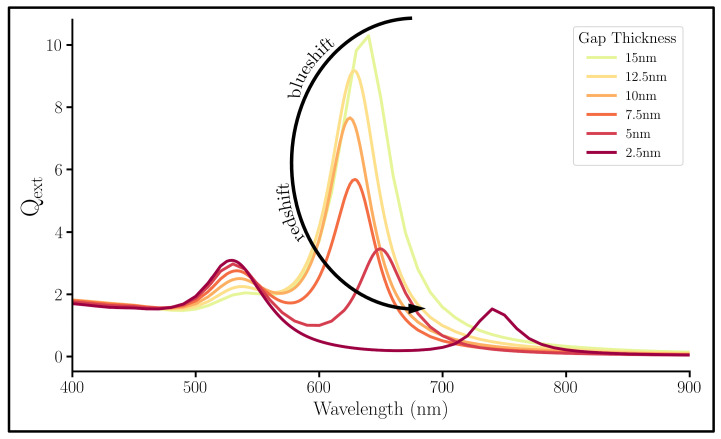
Q${}_{\mathrm{ext}}$ spectra of multilayered NPs with a variable SiO${}_{2}$ gap, a fixed 10 nm Au shell, and a fixed 10 nm radius Au core.

**Figure 6 nanomaterials-13-02893-f006:**
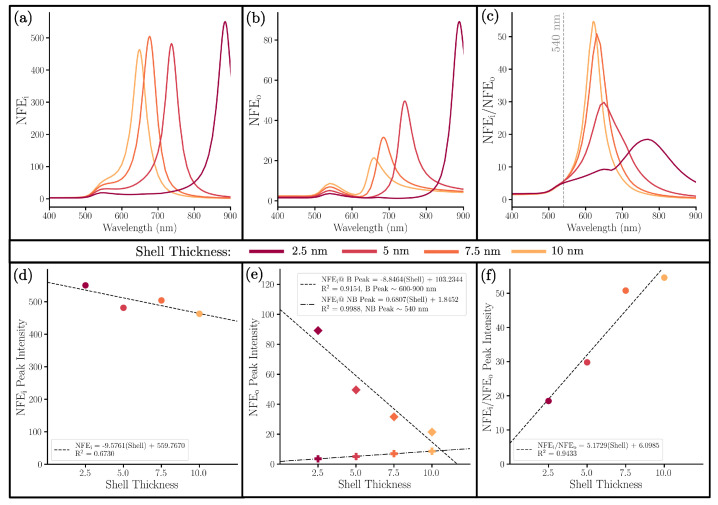
(**a**) NFE${}_{i}$, (**b**) NFE${}_{o}$, and (**c**) NFE${}_{i}$/NFE${}_{o}$ of NMs with a variable Au shell thickness, a fixed 5 nm SiO${}_{2}$ gap, and a fixed 10 nm radius Au core. Below that are correlation plots comparing the shell thicknesses with the intensity of (**d**) the single NFE${}_{i}$ peak, (**e**) both the non-bonding (∼540 nm) and bonding (∼600−900 nm) NFE${}_{o}$ peaks, and (**f**) the single NFE${}_{i}$/NFE${}_{o}$ peak.

**Figure 7 nanomaterials-13-02893-f007:**
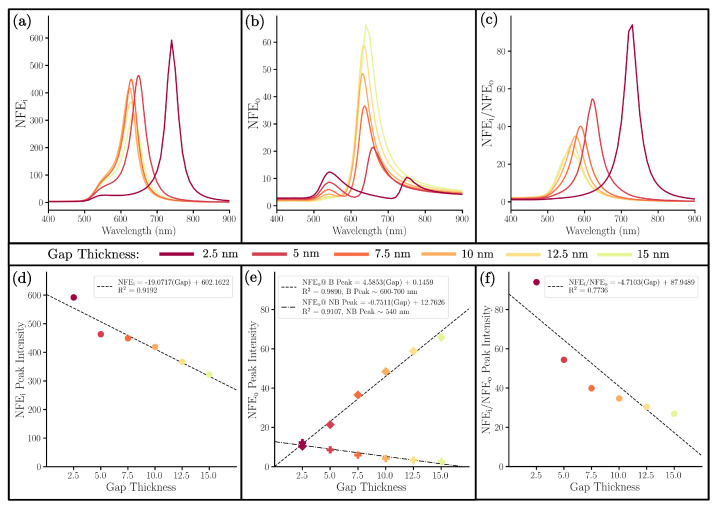
(**a**) NFE${}_{i}$, (**b**) NFE${}_{o}$, (**c**) NFE${}_{i}$/NFE${}_{o}$ of NMs with a variable SiO${}_{2}$ gap thickness, a fixed 10 nm Au shell, and a fixed 10 nm radius Au core. Below that are correlation plots comparing the gap thicknesses with the intensity of (**d**) the single NFE${}_{i}$ peak, (**e**) both the non-bonding (∼540 nm) and bonding (∼600−800 nm) NFE${}_{o}$ peaks, and (**f**) the single NFE${}_{i}$/NFE${}_{o}$ peak.

**Figure 8 nanomaterials-13-02893-f008:**
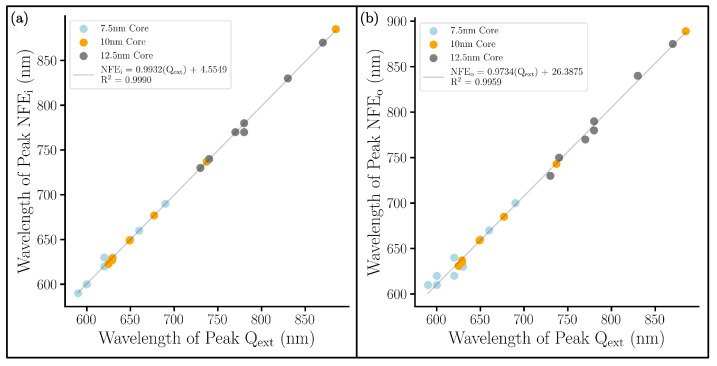
Correlation plots between (**a**) NFE${}_{i}$ and Q${}_{\mathrm{ext}}$ and between (**b**) NFE${}_{o}$ and Q${}_{\mathrm{ext}}$ peaks. All peaks are the bonding peaks between 600 and 900 nm. For each color, a different series of systems were computed, each with a different radius for the core: 7.5 nm (blue), 10 nm (orange), and 12.5 nm (grey).

**Figure 9 nanomaterials-13-02893-f009:**
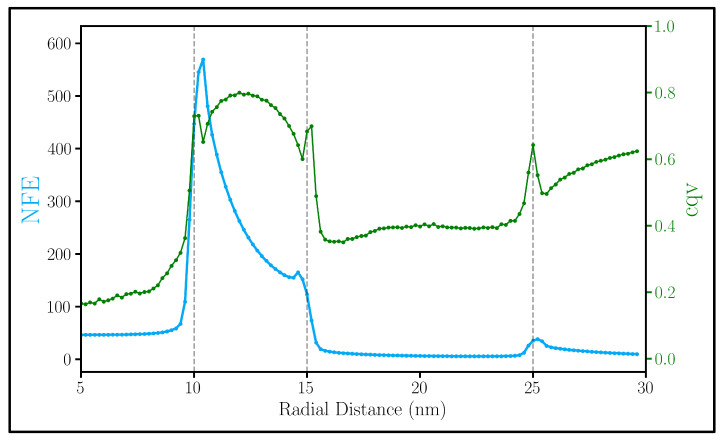
NFE (blue) and cqv (green) as a function of distance from the center of a 10 nm radius Au core, 5 nm SiO${}_{2}$ gap, and a 10 nm Au shell NM at 659 nm. Each point is calculated in 0.2 nm increments. Vertical lines represent the boundaries between different materials.

**Figure 10 nanomaterials-13-02893-f010:**
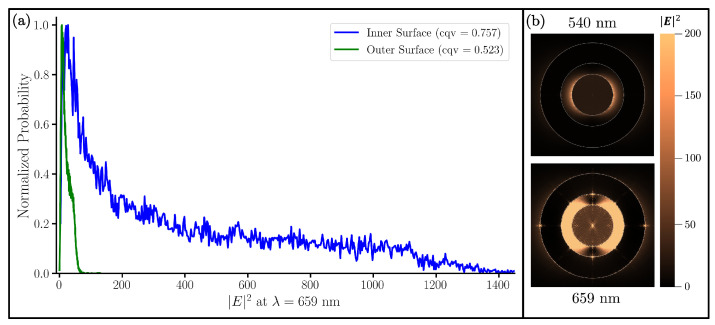
(**a**) Probabilistic histogram of the 10 nm radius Au core, 5 nm SiO_2_ gap, and a 10 nm Au shell NM for measurements of the inner surface (blue) and outer surface (green). Probabilities are normalized by division of the maximum probability of each series. Note that even though the scale of the outer surface is seemingly compressed, its peak spread here is still thinner than the inner surface. (**b**) Visualization of $|E_{\mathrm{inc}}|$ on the same system at peak wavelengths 540 nm and 659 nm.

## Data Availability

The data presented in this study are openly available in Zenodo at https://doi.org/10.5281/zenodo.10056542, accessed on 10 January 2023.

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
