# Peer review of "An Investigation on the Use of Au@SiO2@Au Nanomatryoshkas as Gap-Enhanced Raman Tags"

_nanomaterials, 2023, doi:10.3390/nano13212893_

Round 1
Reviewer 1 Report
Comments and Suggestions for Authors
The paper of Brinton King Eldridge, Saghar Gomrok, Jim Barr, Elise Chaffin, Lauren Fielding, Christian Sachs, Katie Stickels, Paiton Williams, Yongmei Wang, «An Investigation on The Use Of Au/SiO2/Au Nanomatryoshkas as Gap Enhanced Raman Tags» is devoted to the study of Gap-enhanced Raman tags. The influence of gap and shell thickness on extinction spectra and near-field spectra is investigated. It is investigated how the electric field changes in radial direction for layered nanomatrices. It is found that the optimal design for Surface Enhanced Raman Spectroscopy is the combination of thin shell with thin gap.
The topic is of interest to the readers of the journal and the article can be published. However, when reading the article, I have a number of small remarks on the design.
1. The abstract is written as an introduction. In the abstract, the authors try to give some explanation of their results. The abstract should contain only a summary of the results obtained. All explanations are appropriate in the main text.
2. There should be no abbreviations in the abstract.
3. Figure 8 caption should be separated from the main text.
Overall, I think the article is good and should be published.
Reviewer 2 Report
Comments and Suggestions for Authors
Eldridge et al have reported on development of core-shell nanostructure for SERS application. Although it is an interesting topic, there are many issues that need to be addressed forbore suggesting for publication.
- The overall English and structure of this manuscript needs to be improved.
- The introduction does not provide a general overview as to why SERS tag are important and the SERS application, the introduction need to be further strengthened by adding more references (e.g., doi.org/10.1002/adsr.202200039)
- The proposed methodology is only supported by numerical analysis, but it is not put into a test, and it is rather conceptual.
- The authors have not done an experiment to prove the SERS enhancement of proposed geometry using any Raman dye or analyte.
Comments on the Quality of English LanguageModerate editing of English language required
Round 2
Reviewer 2 Report
Comments and Suggestions for Authors
The authors have addressed all the issues raised previously and this revised version is suggested for publication in its current format.